# Low Energy Electron Attachment by Some Chlorosilanes

**DOI:** 10.3390/molecules26164973

**Published:** 2021-08-17

**Authors:** Bartosz Michalczuk, Wiesława Barszczewska, Waldemar Wysocki, Štefan Matejčík

**Affiliations:** 1Faculty of Sciences, Siedlce University, 3 Maja 54, 08-110 Siedlce, Poland; wieslawa.barszczewska@uph.edu.pl (W.B.); waldemar.wysocki@uph.edu.pl (W.W.); 2Department of Experimental Physics, Comenius University, Mlynská dolina F2, 84248 Bratislava, Slovakia; Stefan.Matejcik@fmph.uniba.sk; 3Department of Molecular Physics, National Research Nuclear University MEPhI (Moscow Engineering Physics Institute), Kashirskoe sh. 31, 115409 Moscow, Russia

**Keywords:** dissociative electron attachment, chlorosilanes, thermal electron attachment rate coefficient, activation energy, pulsed Townsend technique

## Abstract

In this paper, the rate coefficients (*k*) and activation energies (*E_a_*) for SiCl_4_, SiHCl_3_, and Si(CH_3_)_2_(CH_2_Cl)Cl molecules in the gas phase were measured using the pulsed Townsend technique. The experiment was performed in the temperature range of 298–378 K, and carbon dioxide was used as a buffer gas. The obtained *k* depended on temperature in accordance with the Arrhenius equation. From the fit to the experimental data points with function described by the Arrhenius equation, the activation energies (*E_a_*) were determined. The obtained *k* values at 298 K are equal to (5.18 ± 0.22) × 10^−10^ cm^3^·s^−1^, (3.98 ± 1.8) × 10^−9^ cm^3^·s^−1^ and (8.46 ± 0.23) × 10^−11^ cm^3^·s^−1^ and *E_a_* values were equal to 0.25 ± 0.01 eV, 0.20 ± 0.01 eV, and 0.27 ± 0.01 eV for SiHCl_3_, SiCl_4_, and Si(CH_3_)_2_(CH_2_Cl)Cl, respectively. The linear relation between rate coefficients and activation energies for chlorosilanes was demonstrated. The DFT/B3LYP level coupled with the 6-31G(d) basis sets method was used for calculations of the geometry change associated with negative ion formation for simple chlorosilanes. The relationship between these changes and the polarizability of the attaching center (α_centre_) was found. Additionally, the calculated adiabatic electron affinities (AEA) are related to the α_centre_.

## 1. Introduction

Studies of the processes of low energy electron attachment by molecules of electronegative gases containing halogen in the gas phase are an important part of current research. They are interdisciplinary, fundamental, and practical. Low-energy electrons can be efficiently captured in the gas phase by many molecules in a two-step process with the formation of an excited negative ion (Reaction 1)
e^−^ + M → M^−*^,(1)
which can be further undergone in the Reactions (2)–(5):M^−*^ → M + e^−^ (autodetachment),(2)
M^−*^ → M^−^ + hν (radiative stabilization),(3)
M^−*^ → M^−^ (internal stabilization),(4)
M^−*^ → R + X^−^ (dissociation),(5)

Stabilization can also be achieved in a collision with a third body (Reaction 6)
M^−*^ + S → M^−^ + S(6)

The formation of a reactive intermediate form as a result of the attachment of electrons to a molecule initiates and drives chemical and physical changes in a wide variety of environments, from interstellar clouds to technological plasmas or even to living tissue [1,2,3,4,5,6,7,8,9]. Knowledge of such reactions, understanding them, and the ability to control them provides new possibilities that can be used in both science and technology.

Silanes and halogen silane derivatives play an important role in plasma technologies that are widely used in the plasma processing industry. In such technologies, the electron attachment process is usually the initial step that ultimately dissociates molecules into radicals and ions. These, in turn, are the components that perform plasma processing. Thus, data on the interaction of electrons with chlorosilanes can be used to control important species in plasma in many technologies, especially in the microelectronics industry. The modern microelectronics industry is largely dependent on plasma-assisted processes such as cleaning, etching, and deposition [10,11,12].

In previous articles in this series [13,14], we focused on the importance of quantitative data regarding the process of electron attachment by the chlorosilanes. These data are necessary for both understanding the basic processes in plasma and for modelling the plasma processing procedure [15]. In our continued efforts to provide such data, this article focuses on SiHCl_3_, SiCl_4_, and Si(CH_3_)_2_(CH_2_Cl)Cl. To the best of our knowledge, there are no kinetic data available regarding the electron attachment to SiHCl_3_, SiCl_4_, and Si(CH_3_)_2_(CH_2_Cl)Cl. There are only two articles [13,14] that report on the electron attachment rate coefficient (*k*) and the activation energy (*E_a_*) for chlorosilanes. They come from our laboratory and concern seven chlorosilanes: SiCH_3_Cl_3_, SiH(CH_3_)_2_Cl, SiHCH_3_Cl_2_, Si(C_2_H_5_)_3_Cl, Si(CH_3_)_3_Cl, Si(CH_3_)_2_Cl_2_, and SiH(CH_3_)_2_(CH_2_Cl). In turn, the formation of negative ions following low-energy electron attachment to chlorosilanes was investigated incrossed electron–molecular beam experiments applying mass spectrometric detection to the anions. In particular, this refers to works about SiCl_4_ [16,17,18,19,20,21,22,23]; SiHCl_3_ [18]; SiH_2_Cl_2_ [18]; (CH_3_)_3_SiCl; and Ph(CH_3_)_2_SiCl [24] (Ph refers to phenyl group). Other works report data for the chlorinated vinyl derivatives of silane, i.e., SiCl_3_C_2_H_3_, Si(C_2_H_3_)_4_, Si(CH_3_)_3_C_2_H_3_ (Kočišek et al. [25]), cyclo-C_5_H_10_SiCl_2_ and cyclo-C_5_H_10_SiH_2_ (Kumar et al. [19]), and the chlorinated cyclic silane derivatives 1,1-dichloro-1-silacyclohexane and silacyclohexane (Bjarnason et al. [26]).

For several years, we have been conducting systematic research in our laboratory to study the mechanism and kinetics of low-energy electron attachment processes by various groups of molecules. Our interest in these types of processes results from both practical and cognitive aspects. The motivation for the research concept on the group of halogenated silanes was the fact that the literature lacked experimental kinetic data for that group of compounds, which is so important for microelectronics. Moreover, there was also no systematic research in one laboratory that would have made it possible to find a relationship between the structure of these molecules and their electron capture ability.

This work is a continuation of our previous research [13,14] in which we presented (for the first time) kinetic data for thermal electron attachment using some chlorinated derivatives of silane obtained by means of a time-resolved electron swarm method. In this article, we present the results of a study on low-energy electron attachment by SiHCl_3_, SiCl_4_ and Si(CH_3_)_2_(CH_2_Cl)Cl (for the structures of all investigated molecules see Figure 1).

Using the Gaussian 16 software package at the DFT/B3LYP level coupled with the 6-31G(d) basis sets method, we have calculated the changes in a molecular structure when going from a neutral to a transient negative ion and adiabatic electron affinity (AEA) for SiCl_4_, SiHCl_3_, SiH_2_Cl_2_, and SiH_3_Cl. The results have been compared with the *α*_center_ of these molecules. Additionally, bond dissociation energies (BDE) for a Si-Cl bond were calculated.

## 2. Results and Discussion

Here, we report the swarm results of thermal electron attachment processes to three chlorosilanes (SiHCl_3_, SiCl_4_ and Si(CH_3_)_2_(CH_2_Cl)Cl) for the first time. Thermal electron attachment rate coefficients (*k*) and activation energies (*E_a_*) were measured using the drift tube setup, which involves a pulsed Townsend (PT) technique. The rate coefficients were measured at nine temperatures (*T*) ranging from 298 to 378 K (where *T* means both the electron temperature *T_e_* and the gas temperature *T_g_*). Carbon dioxide was used as a buffer gas, which quickly thermalized the electron swarm. The applied concentration of the electron attaching gases (chlorosilanes) depended on their efficiency in attaching electrons and was chosen to produce a process rate of approximately 10^5^ s^−1^. In our experiment, the disappearance of the electrons from the swarm was monitored. The rate coefficients that were obtained correspond to the total attachment processes, including all of the reaction channels.

For all of the investigated molecules, we observed that the electrons were attached only in the two-body processes. The example results (at 298 K) for SiHCl_3_ in terms of the rate of electron disappearance (V_−e_) from the swarm vs. SiHCl_3_ concentration for three different concentrations of CO_2_ as a diluting gas are shown in Figure 2. The very good straight lines as well as zero intercepts show that for the case of SiHCl_3_ as well as for the other ones that we investigated (Table 1), the rate of electron disappearance depends only on SiHCl_3_ concentration. This means that only two-body process electron attachment by a single molecule occurs.
SiC_n_H_m_Cl_x_ + e^−^ → products(7)

All of the presently obtained rate coefficients at room temperature (*T* = 298 K) are collected in Table 1 together with our recent data from the PT technique and with available literature data that are necessary for further discussion. The data obtained in this work were determined from five measurements. The corresponding rate coefficients at 298 K are equal to (5.18 ± 0.22) × 10^−10^ cm^3^·s^−1^, (3.98 ± 1.8) × 10^−9^ cm^3^·s^−1^, (8.46 ± 0.23) × 10^−11^ cm^3^·s^−1^ for SiHCl_3_, SiCl_4_, and Si(CH_3_)_2_(CH_2_Cl)Cl, respectively. To our knowledge, for all of the molecules, there are so far no literature data available for the electron attachment rate coefficient.

We have also measured the rate coefficients *k*(*T*) in the temperature range 298–378 K. The results for the investigated systems in terms of ln(*k*) vs. 300 K/T are shown in Figure 3. In all cases, we observed an increase in rate coefficients with temperature according to the Arrhenius equation, *k(T) = k*_1_
*exp[−E_a_/(k_B_T)]*, where *k*_1_ is the preexponential factor, *k_B_* is the Boltzmann constant, and *E_a_* is the activation energy. This means that the studied processes require certain activation energies, and *E_a_* is derived from the fits the experimental data *k*(*T*). The accuracy of the Arrhenius equation has been studied theoretically by Fabrikant and Hotop [27]. They demonstrated that in both exo- and endothermic reactions, the equation is valid in a finite range of intermediate temperatures. Since our experiments were conducted in the temperature range in which, according to the above calculations, the Arrhenius law applies, and because we observe a clear linear relationship ln(*k*) vs. 300 K/T, *E_a_* can then be obtained from the slope of the curves in Figure 3. The values of *E_a_* for the measured chlorosilanes are collected in the Table 1 column for *E_a_*. The activation energies are equal to 0.25 ± 0.01 (eV), 0.20 ± 0.01 (eV), and 0.27 ± 0.01 (eV) for SiHCl_3_, SiCl_4_, and Si(CH_3_)_2_(CH_2_Cl)Cl, respectively. All of the presented values were determined for the first time, which is similar to the rate coefficients.

The activation energy is identified with the point of intersection of the potential curves of a neutral molecule and an ion (Figure 4), where the resonances are of low energy. However, due to the tunnelling effect that can occur from the vibration level below the barrier, it is likely that the activation energy of the process is slightly lower than the intersection energy. The activation energy thus determines the effective barrier energy for the electron attachment process.

Data for some chlorosilanes in the gas phase obtained using the crossed electron-molecular beam technique are available in the literature. These experimental studies of the interaction of low-energy electrons with chlorosilanes mainly concern the formation of negative ions or the determination of cross sections for electron attachment. Wan et al. [18] investigated the processes of dissociative electron attachment (DEA) by chlorosilanes. These studies provide data for SiHCl_3_, SiCl_4_, and SiH_2_Cl_2_. They measured the cross sections for dissociative electron attachment at the detection threshold of 2 × 10^−18^ cm^2^. The experiment demonstrated that only the cross section for SiHCl_3_ exceeded this value. Its maximum occurs at an electron energy of 1.2 eV. The shift of the peak for dissociative electron attachment towards lower energy relative to the electron attachment peak is a common phenomenon related to the dependence of the lifetime of the on the reciprocal of the electron energy [36,37,38]. Dissociative attachment at a threshold electron energy (0 eV) is not possible because the Si-Cl bond energy exceeds the electron affinity of the chlorine atom. The observations suggest that the lifetime of the parent ion is in the order of 1 μs [18].

One of the latest papers dealing with SiCl_4_ and its interactions with electrons is the work by Kumar et al. [19]. The formations of negative ions have been measured in the energy range from 0 to 10 eV. They have presented a complete and consistent picture of electron attachment to this molecule, and they have offered an explanation for the inconsistency between the previous studies (conducted by various research groups [16,18,20,21,22,23]). These studies showed that electron attachment to SiCl_4_ leads to the formation of the fragment anions: Cl^−^ located at 1,8 eV and 7.1 eV; Cl_2_^−^ located at 7.7 eV; SiCl_2_^−^ located at 7.7 eV; and SiCl_3_^−^ located at 2.1 eV and 6.9 eV. Additionally, the molecular anion SiCl_4_^−^, was observed with appreciable intensity at 0 eV. No fragmentation was observed at such low energies. In turn, Böhler and co-workers [24] analysed the dissociative electron attachment reactions by chlorotrimethylsilane (Me_3_SiCl) and chlorodimethylphenylsilane (PhMe_2_Cl) in the electron energy range of 0–20 eV to show the possible effect of an aromatic side group on the dissociation of a Si-Cl bond. These studies showed that Cl^−^ appears as the dominant negative fragment (between 5.5 eV and 12 eV with a maximum at 7 eV). Additionally, the Cl^−^ signal was observed at about 1 eV, but these signals were attributed to impurities. Moreover, it has been shown that DEA also cleaves the Si-C bonds, leading to loss of the neutral or anionic side groups -CH_3_ and -C_6_H_5_ and also to the cleavage of the C-H bond with the consequent loss of the neutral hydrogen atom. Moreover, for the Me_3_SiCl molecule, the appearance of the CH_3_Cl^−^ ion also proves the reorganization of the bond.

Similar observations were made by Kočišek et al. [25], who analysed the formation of negative ions by electron attachment by vinyl silane derivatives: trichlorovinylsilane (TCVS, SiCl_3_C_2_H_3_), tetravinylsilane (TVS, Si(C_2_H_3_)_4_), trimethylvinylsilane (TMVS, Si(CH_3_)_3_C_2_H_3_), and trimethoxyvinylsilane (TMOVS, Si(CH_3_O)_3_C_2_H_3_). The research was conducted in the energy range of 0–12 eV. They showed that DEA to TCVS and TVS leads to the formation of a wide variety of ionic fragments. These reactions range from simple bond cleavages to much more complex single-molecular cleavages including multiple bond cleavages, hydrogen transfer in a temporary negative ion, and the formation of new bonds. They found that all of the analysed silanes together with trichlorovinylsilane (TCVS) are weak electron acceptors.

Such observations strongly contrast with the behaviour of the hydrocarbon analogues, in which chlorination, in most cases, leads to a significant increase in the DEA cross-sections, resulting in a very strong signal, which usually appears in a narrow range just at the threshold (0 eV) [39]. In the case of chlorosilanes, DEA shows moderate Cl^−^ signals at higher energies. This different behaviour of chlorosilanes can be attributed to different energetics of the corresponding DEA reaction, which, in turn, is the result of a stronger Si-Cl binding energy compared to C-Cl. The experimental dissociation energies (BDE) for CCl_4_ (C-Cl bond) and for CHCl_3_ (C-Cl bond) are 2.99 eV [40,41] and 3.22 eV [42], respectively. In turn, for SiCl_4_ (Si-Cl bond), BDE = 4.83 eV [42], and for SiHCl_3_, BDE = 4.05 eV [42]. This is also confirmed by our theoretical calculations at the DFT/B3LYP level coupled with the 6-31G(d) basis sets method. The obtained BDE values (C-Cl, Si-Cl) are 2.51 eV, 2.85 eV, 4.18 eV, and 4.28 eV for CCl_4_, CHCl_3_, SiCl_4_, SiHCl_3_, respectively.

Studies in our laboratory also confirmed the different chlorosilane behaviour. In our previous works [13,14], we presented the kinetic results on thermal electron attachment by some chlorosilanes (SiCH_3_Cl_3_, SiH(CH_3_)_2_Cl, SiHCH_3_Cl_2_ and Si(C_2_H_5_)_3_Cl, Si(CH_3_)_3_Cl, and Si(CH_3_)_2_Cl_2_ and SiH(CH_3_)_2_(CH_2_Cl)). We showed that the rate coefficients at 298 K for these molecules were of the same order of magnitude: 8.72 × 10^−11^ cm^3^·s^−1^, 6.72 × 10^−11^ cm^3^·s^−1^, 16.8 × 10^−11^ cm^3^·s^−1^ and 3.27 × 10^−11^ cm^3^·s^−1^, 9.56 × 10^−^^11^ cm^3^·s^−^^1^, 6.62 × 10^−^^11^ cm^3^·s^−^^1^, 1.24 × 10^−^^11^ cm^3^·s^−^^1^ for SiCH_3_Cl_3_, SiH(CH_3_)_2_Cl, SiHCH_3_Cl_2_, Si(C_2_H_5_)_3_Cl, Si(CH_3_)_3_Cl, Si(CH_3_)_2_Cl_2_ and SiH(CH_3_)_2_(CH_2_Cl), respectively (Table 1). It was an unusual observation and strongly contrasted with the general behaviour of chloroalkanes. Earlier experimental and theoretical studies conducted in our laboratory [31,35,43,44,45,46] on the group of chloroalkanes clearly showed that the value of the rate coefficient of the electron attachment process is influenced by the number of chlorine atoms as well as their arrangement in the carbon chain. For example, the analysis of kinetic data for chlorine derivatives of methane showed that the appearance of another chlorine atom in the chlorine derivative molecule significantly influences the value of the rate coefficient of the electron attachment process. The electron attachment reaction rate coefficient s (*k*) of this group of compounds are CCl_4_ (*k* = 3.9 × 10^−^^7^ cm^3^·s^−^^1^ [30]); CHCl_3_ (*k* = 3.7 × 10^−^^9^ cm^3^·s^−^^1^ [29]); CH_2_Cl_2_ (*k* = 1.8 × 10^−^^13^ cm^3^·s^−^^1^ [28]); and CH_3_Cl (*k* = 1.1 × 10^−^^17^ cm^3^·s^−^^1^ [28]) (Table 1). As we can see, each successive chlorine atom substituted for hydrogen increases the value of the rate coefficient by several orders of magnitude. The largest change is visible at the transition from CCl_4_ to CH_3_Cl, where the rate coefficient is increased by ten orders of magnitude. If we compare the values of the *k* for molecules with the same number of chlorine substituents but in a different position, e.g., CH_2_ClCHCl_3_ (*k* = 3.1 × 10^−^^7^ cm^3^·s^−^^1^ [35]) or CHCl_2_CHCl_2_ (*k* = 3.5 × 10^−^^8^ cm^3^·s^−^^1^ [35]) (Table 1), we notice differences in the *k* values, this time, by an order of magnitude. In the case of the previously investigated chlorosilanes containing the -CH_3_ and -C_2_H_5_ groups, we did not observe such relationships.

Due to the lack of literature data, the results (*k, E_a_*) obtained in this work will be compared with the kinetic data for other molecules of a similar structure. The analysis from Table 1 shows that the rate coefficients (at 298 K) for SiCl_4_ (3.98 × 10^−9^ cm^3^·s^−^^1^) are two orders of magnitude lower than they are for CCl_4_ (3.9 × 10^−7^ cm^3^·s^−^^1^). The activation energies vary from 0.20 eV (present data) for SiCl_4_ to 0 eV for CCl_4_. On the other hand, when comparing the rate coefficients for SiHCl_3_ (5.18 × 10^−10^ cm^3^·s^−^^1^) and CHCl_3_ (3.7 × 10^−9^ cm^3^·s^−^^1^), it can be seen that the *k* for SiHCl_3_ is one order of magnitude lower than for CHCl_3_. Activation energies vary from 0.25 eV (present data) to 0.11eV for SiHCl_3_ and CHCl_3_, respectively. The rate coefficient for Si(CH_3_)_2_(CH_2_Cl)Cl (such as in our previous works [13,14]) is of the order of 10^−11^ and is equal to 8.46 × 10^−11^ cm^3^·s^−^^1^ and *E_a_* = 0.27 eV. A few comments can be made after this inspection. Chlorosilanes are weaker electron scavengers than the corresponding chloroalkanes, as reflected in the values of the rate coefficients and activation energies. In the case of simple chlorosilanes (SiHCl_3_, SiCl_4_), the influence of the number of chlorine atoms on the *k* value and thus on the *E_a_* value is observed. Such relationships were not observed for molecules with a complex structure, such as Si(CH_3_)_2_(CH_2_Cl)Cl.

In order to verify the obtained research results, theoretical analysis of the influence of the molecule structure on the efficiency of the electron attachment process was performed. Simple dependencies of the kinetics of the electron attachment process on some molecular parameters work well for a specific group of compounds (this has been confirmed, for example, for chloromethanes [43,44]). In this article, a group of molecules, SiH_4-*n*_Cl_*n*_ (*n* = 0–4), have been analysed. The influence of such molecular parameters as dipole moment (*μ*), molecule polarizability (*α*), electron polarizability of the attaching center (*α*_center_), relative change in the bond length in the molecule after electron attachment (Δ*r/r*), relative change in the equilibrium angle (Δ*∠**/**∠*), and adiabatic electron affinity (AEA) were investigated.

Our analysis showed that the electron attachment efficiency does not depend on the dipole moment (*μ*) of the molecule for the two-body process (reaction 7), for example, comparing the rate coefficients of the electron attachment process by the molecules characterized by a zero dipole moment, e.g., CCl_4_ (*k* = 3.9×10^−7^ cm^3^ s^−1^ [30]); SiCl_4_ (*k* = 3.98 × 10^−9^ cm^3^·s^−1^, present data); or CF_4_ (*k* < 10^−16^ cm^3^·s^−1^ [33]). As we can see, CCl_4_ is one of the most efficient electron scavengers, while CF_4_ captures electrons extremely slowly. These statements are also confirmed by the research of Dashevskaya et al. [47] who, on the basis of the Vogt–Wannier [48] theory and its extension by Fabrikant and Hotop [49] have provided analytical approximations for the cross sections and thermal rate coefficients for the attachment of electrons by polar and polarizable molecules. Their calculations for many molecules showed that the impact of the dipole or quadrupole moment on the efficiency of the electron attachment process is small.

In our analyses we also considered the influence of the polarizability of the molecule on the efficiency of the electron attachment process. Klar et al. [50] showed that the *α* can be an important factor in determining the value of the thermal electron attachment rate coefficient. They presented the relationship between *k* and *α* for the thermal electron attachment rate, described by the equation *k* = 4 × 10^−8^ α^1/2^ cm^3^·s^−1^, where the *α* values are expressed in 10^−24^ cm^3^ units. However, as our analyses have shown, this relationship is only valid for highly halogenated halocarbons, for which the thermal s-wave electron cross-section approaches its limiting value *πλ*^2^, where *λ* is the reduced de Broglie wave length. For other molecules, a stronger dependence is observed than it does for the results from the equation proposed by the researchers.

Both the experimental and theoretical research conducted in our laboratory [43,44] (concerning the group of halogenated derivatives of methane) showed that the efficiency of the electron attachment process does not depend on the polarizability of the molecule as a whole but on the *α*_center_ of the molecule. As an attaching center, we consider the part of the molecule that is immediately connected to the attachment process, e.g., in the case of CH_n_-X_m_ (where X are halogen atoms), the *α*_center_ is a sum of the polarizabilities of the halogen atoms calculated using an additivity rule, while carbon and hydrogen atoms are eliminated from consideration. These data confirm that in the case of the two-body electron attachment process (for a group of molecules substituted with the same halogen atom), ln(*k*) and *E_a_* are straight-line functions of *α*_center_. They also confirmed that the increase in *α*_center_ decreases the *E_a_* of the electron attachment process, and thus increases the *k* of this process. Moreover, the *α*_center_ determines changes in the equilibrium bond length due to the formation of the negative ion. We also expected similar effects for the group comprising simple chlorosilanes SiH_4-*n*_Cl_*n*_ (*n* = 0–4). Theoretical calculations of changes in the structure of the molecule related to electron attachment were made. Studies started with SiH_3_Cl. In Figure 4, potential curves for a SiH_3_Cl molecule and its negative ion are presented.

The validity of the calculations can be estimated by the comparison of the obtained parameters with the available experimental values: the equilibrium Si–Cl bond length, r(Si–Cl) = 2.08 cm^−8^ (calc.) and 2.05 cm^−8^ (exp.) [51]; the Si-Cl bond energy, BDE(Si–Cl) = 4.28 eV (calc.) and 4.75 eV (exp.) [42], the chlorine atom electron affinity, EA(Cl) = 3.5 eV (calc.) and 3.6 eV (exp.) [42] as well as the H–Si–Cl angle, ∠HSiCl = 108.6° (calc.) and 108.3° (exp.) [51]. The calculated values are not far from the experimental ones, and it seems that they can be used safely for comparison purposes.

As a result of the strong electrostatic interaction of an additional electron occupying LUMO, the (Si-Cl^−^) bond becomes elongated and weakened (the depth of the potential well decreases), and the energy of its equilibrium position in relation to Si-Cl changes. Table 2 and Figure 5 show the calculated molecular parameters for SiH_3_Cl, SiH_2_Cl_2_, SiHCl_3_, SiCl_4_ molecules and the corresponding ions. Figure 5 shows the potential curves for simple chlorosilanes and its negative ion. As it can be seen from the presented table (Table 2) and Figure 5, as expected, the equilibrium internuclear distance for the Si–Cl bond in the molecule, r(C–Cl), does not change appreciably, while the well depth diminishes slightly from SiH_3_Cl to SiCl_4_. The main changes occur when the ion is formed: r(Si–Cl^−^) shortens significantly with the number of Cl atoms, and the adiabatic electron affinity, AEA, increases. As a result of elongating the bond, the point of intersection of the potential curves of the molecule and the ion shifts towards higher energies, which causes an increase in the *E_a_* of the electron attachment process and a decrease in the value of the *k*.

Table 2 and Figure 6 show the relationship between the relative elongation of the Si-Cl bond and the electron polarizability of the attaching center (*α*_center_) for SiH_3_Cl, SiH_2_Cl_2_, SiHCl_3_, and SiCl_4_. The *α*_center_ was calculated as *α*_center_ = N_Cl_ × α_Cl_, where N_Cl_ is the number of chlorine atoms, α_Cl_ is the chlorine atom polarizability, and the α_Cl_ = 2.18 × 10^−24^ cm^3^ [42]. The obtained data clearly show a significant influence of the *α*_center_ on the degree of binding elongation. The change in the bond length is greater the smaller the *α*_center_ is. The smallest change is observed for SiCl_4_ (0.110), and the change is observed largest for SiH_3_Cl (0.249) (see Table 2). This influences the activation energy of the process and thus the rate coefficient as observed in our experiment (*E_a_* = 0.20 eV, *k* = 3.98 × 10^−9^ cm^3^·s^−1^ and *E_a_* = 0.25 eV, *k* = 5.18 × 10^−10^ cm^3^·s^−1^ for SiCl_4_, SiHCl_3_, respectively).

Using the same method, we have also calculated the adiabatic electron affinity (AEA). Adiabatic electron affinity is the difference between the energy of the molecule (E_tot_) at its most stable geometry and of the negative ion when it is also at its most stable conformation (AEA (M) = E_tot_ (M) – E_tot_ (M^−^)). The obtained data are collected in Table 2 (column AEA). Figure 7 shows this data as a function of *α*_center_. Accordingly with this graph, AEA increases with increasing *α*_center_. These are the preliminary data. They give some correlation between the molecular structure and its ability to capture an electron. Of course, to achieve more detailed insight, many more calculations are necessary (for the larger group of chlorosilanes).

The dependence of an attachment cross-section on energy has a resonant form which is a result of a Franck–Condon transition from the ground vibrational level of the neutral to the transient negative ion. The energy of this transition is the vertical attachment energy (VAE) in Figure 4. In our previous papers [35,52], we have also shown that there is a strong correlation between the *α*_center_ values for the electron attachment and VAE value measured by Aflatooni et al. [53]. We have observed good linear dependence for two groups of compounds—chloroalkanes and chlorofluoroalkanes. Experimental literature data for VAE are available for the methyl derivatives of silanes and for SiCl_4_ [54]. Due to the lack of VAE values for a series of simple chlorinated silanes, we are not able to present the relationship between VAE vs. *α*_center_.

We have also observed some changes in the equilibrium angle as a result of the ion formation, which can cause an additional energetic barrier in this process. These data are presented in Table 2. As you can see, this effect also diminishes with the increasing *α*_center_.

The main observed features for the thermal electron attachment dependence on the molecular parameters are following:

The effectiveness of the electron capture does not depend on the dipole moment of the molecule for two-body process. It also does not depend on the overall electronic polarizabilities. The polarizability of the attaching center determines the change in the equilibrium bond length caused by the formation of the negative ion. This, in turn, determines the position of the intersecting point of potential curves (Figure 5) for the molecules and their negative ions. A practical conclusion is that the expected rate coefficients for many compounds can easily be estimated, or the measured ones can easily be verified on the basis of the *α*_center_.

Furthermore, to judge the quality of the experimental results, if both the rate coefficient and activation energy are well known, we can use the theoretical relationship between *k* and *E_a_* presented in Figure 8 (solid line) and discussed in detail in our previous works [55,56]. The pre-exponential factor *A* in the Arrhenius equation was interpreted as the maximum value of the electron attachment rate coefficient for strongly attaching molecules with zero activation energy (such as CCl_4_) at room temperature (*k* = 3.79 × 10^−7^ cm^3^ s^−1^ [57]). According to the postulates of quantum mechanics, the electron can be viewed as the de Broglie wave. Since the maximum s-wave cross section of electron scattering can be described by the expression *λ**^2^/4**π**,* where *λ* is the de Broglie wavelength for the electron of energy *3/2k_B_T*, the factor *A* can be treated as the theoretical limit for the rate coefficient 5.0 × 10^−7^ cm^3^ s^−1^ at 298 K. This value of *A* is very close to the thermal rate coefficient presented for the CCl_4_ molecule. The Arrhenius theoretical relationship was determined based on the experimental data for the chlorinated derivatives of alkanes, and as was stated, is also fulfilled for the chlorinated and fluorinated alcohols [58,59], perfluoroethers [60], and some chlorosilanes [13,14].

In Figure 8, *k* is presented as a function of the *E_a_* for chlorinated silanes (present data) together with experimental data for chlorinated silanes and chlorinated alkanes (Table 1). As it is shown in Figure 8, most of the experimental data are located practically nearby the theoretical line. Such location of the points (*E_a_, k*) proves that the activation energy is the main factor determining the rate coefficient for thermal electron attachment. This can offer a good criterion for judging the quality of the experimental results. Additionally, the experimental kinetic results are much more reliable if both kinetic parameters, i.e., *k* and *E_a_,* are determined in the same experiment.

## 3. Materials and Methods

### 3.1. Chemicals

The investigated compounds and their purities were as follows: tetrachlorosilane (SiCl_4_, Aldrich, St. Louis, MO USA, ≥99%), trichlorosilane (SiHCl_3_, Aldrich, St. Louis, MO USA, ≥99.5%) and Chloro(chloromethyl)dimethylsilane (SiCl(CH_3_)_2_(CH_2_Cl), Aldrich, United Kingdom, 99%). Before starting the experiment, all samples were degassed using a freeze–pump–thaw method at the temperature of liquid nitrogen. As a buffer gas, carbon dioxide (from Fluka, Poland) was used with a stated purity of 99.998% as delivered.

### 3.2. The Swarm Technique

The Swarm technique, also known as the pulsed Townsend technique, was applied for the measurements of the electron attachment rate coefficients and activation energies. This technique relies on the measurement of the current generated by a traversing electron swarm through the gap between parallel electrodes to which a uniform electric field has been applied. Whole experimental procedure has been described in detail previously [55]. Thus, a brief description will be given.

The electron swarm is generated by a 5 ns laser (Nd:YAG operating on fourth harmonics, 266 nm, 10 Hz). The laser beam is focused after it passes through the optical system and then through the quartz window; it enters the reaction chamber and finally incidents at the surface of the stainless steel cathode. Liberated photoelectrons move towards the anode and interact with the gas molecules. During that passage, ions are also produced, but their velocity compared to that of the electrons is slow. In the present study, carbon dioxide was used as a buffer gas, and in those particular conditions, photoelectrons were in thermal equilibrium with the gas molecules.

Electrons moving in the homogenous electric field influence the potential of the collecting electrode. This change in anode potential is different and depends on whether the electron scavenger is present or not. In the pure carbon dioxide, anode potential grows linearly as the electron swarm traverses the gap between electrodes. This linear dependence is not observed when additives are introduced into the chamber. Additives that are better electron scavengers result in a stronger deviation from the linear relationship. The output signal is amplified and after it is registered on the oscilloscope and finally is saved in the PC memory. From the shape of the output signal, the electron attachment rate coefficients are determined.

To prepare the scavenger–CO_2_ mixture, first, the studied gas is introduced into the chamber and the buffer gas is then injected to the total pressure of the mixture in the range 4.7 × 10^4^–5.3 × 10^4^ Pa (350–400 Torr). The experiments were conducted for a few different scavenger concentrations. For each density, reduced electric field value (E/N) 50 pulses were averaged and analysed. The experiment was performed for few E/N values in the range of 1.5–3.0 Td (1 Townsend = 10^−17^ V cm^2^).

### 3.3. Calculations

All calculations presented in this work were performed using the Gaussian 16 software package [61] at the DFT/B3LYP level [62,63] coupled with the 6-31G(d) basis sets. The geometries of the neutral molecules and their anions were fully optimized, and calculated energies were corrected for zero-point energy (ZPE). Bond dissociation energies (BDE) were calculated as the difference between the enthalpy of the neutral molecule and the sum of the enthalpies of the two radicals formed after breaking the bond in this molecule. The potential energy surfaces (PES) for the Si–Cl bond in the neutral molecules of the compounds and their anions were used to determine the adiabatic electron affinities (AEA).

## 4. Conclusions

Results from the study on thermal electron attachment by SiCl_4_, SiHCl_3_, and Si(CH_3_)_2_(CH_2_Cl)Cl in the gas phase are reported. The rate coefficients and the activation energies were determined, and all of them were reported and determined for the first time. The measurements were performed in the temperature range of *T* = 298–378 K (*T = T_e_ = T_G_*). It was found that all of the investigated compounds only attach to electrons in a two-body process. The determined rate coefficients for thermal electron attachment processes show an Arrhenius-type rise with increasing temperature. The rate coefficients at 298 K are equal to (3.98 ± 1.8) × 10^−9^ cm^3^·s^−^^1^, (5.18 ± 0.22) × 10^−^^10^ cm^3^·s^−^^1^, and (8.46 ± 0.23) × 10^−^^11^ cm^3^·s^−^^1^, and the activation energies are (0.20 ± 0.01) eV, (0.25 ± 0.01) eV, and (0.27, 0.01) eV for SiCl_4_, SiHCl_3_, and Si(CH_3_)_2_(CH_2_Cl)Cl, respectively. In seeking the link between the rate of the two-body electron attachment in the gas phase and the structure of an accepting molecule, we have analysed the dependence of the rate coefficients on the dipole moments, the overall and electronic polarizability of the molecules as well as those of the attaching center. We have found that the effectiveness of the electron attachment does not depend on the dipole moment or on the polarizability of the molecule as a whole, but on the *α*_center_ of the molecule. Using the Gaussian 16 software package at the DFT/B3LYP level coupled with the 6-31G(d) basis sets method, we have calculated the geometry changes associated with negative ion formation and AEA for SiCl_4_, SiHCl_3_, SiH_2_Cl_2_, and SiH_3_Cl. We have noticed that the changes during bond elongation, some change in the equilibrium angle and AEA depend on the *α*_center_. Additionally, the linear relation between *k* and *E_a_* for chlorosilanes was demonstrated.

Much more experimental data on the *k* and *E_a_* as well as on beam data for the energy dependence of the electron attachment cross-section are required to go into detailed analysis and to also confirm the reliability of the theoretical calculations.

## Figures and Tables

**Figure 1 molecules-26-04973-f001:**
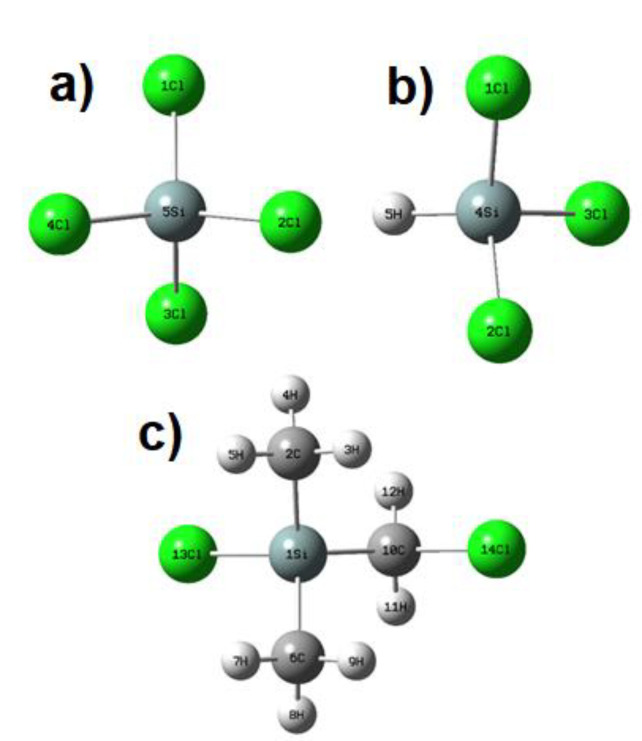
Structures of all investigated chlorinated silanes: (**a**) tetrachlorosilane, (**b**) trichlorosilane, (**c**) chloro(chloromethyl)dimethylsilane.

**Figure 2 molecules-26-04973-f002:**
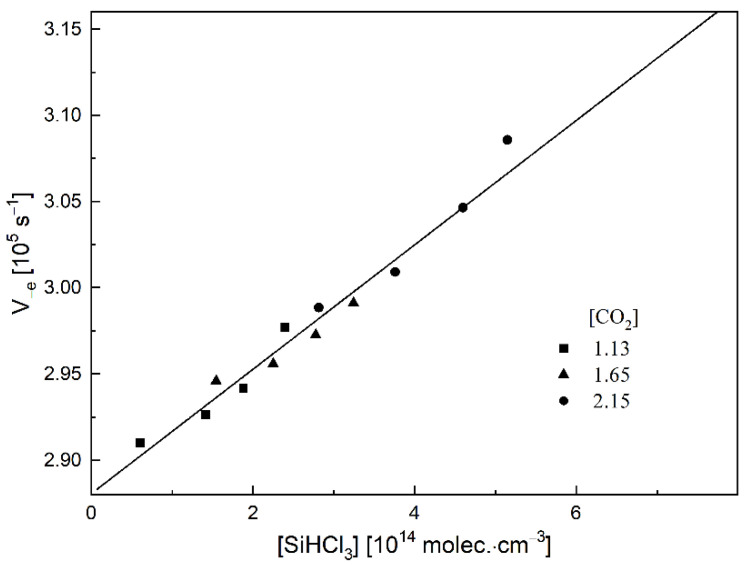
The rate of electron disappearance for SiHCl_3_ as a function of halogenated silane concentration at a three different CO_2_ concentrations [10^19^ molec.·cm^−3^].

**Figure 3 molecules-26-04973-f003:**
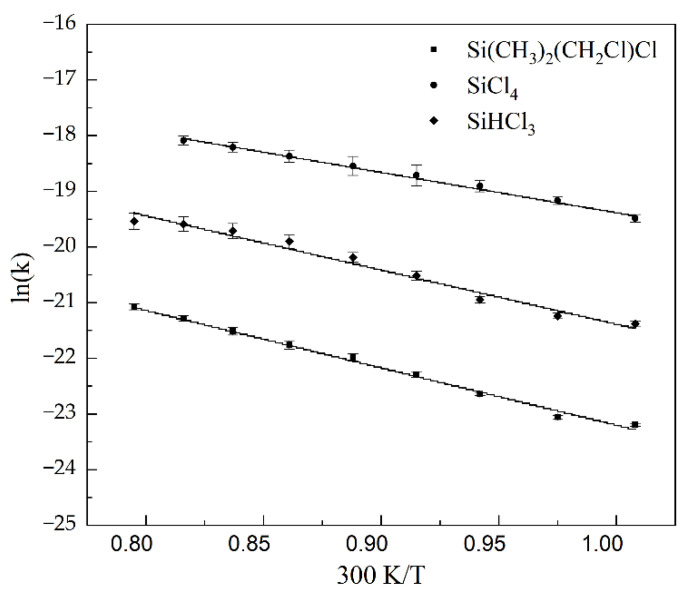
Dependence of the logarithm of *k* vs. 300 K/T for investigated molecules.

**Figure 4 molecules-26-04973-f004:**
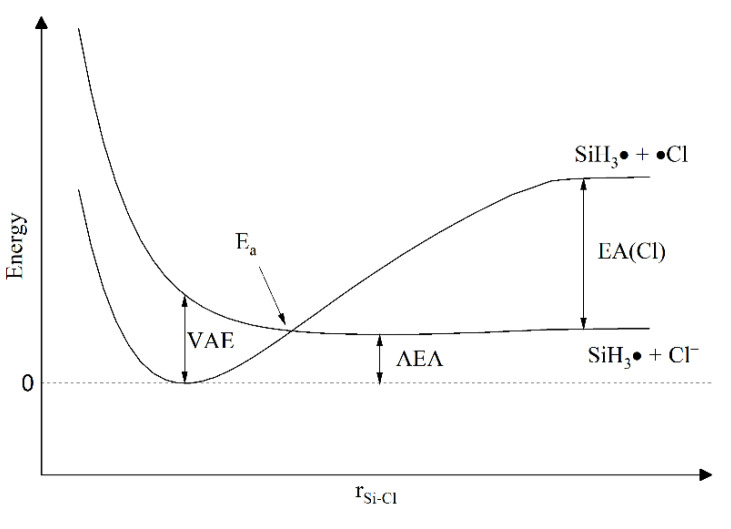
Potential curves for a SiH_3_Cl molecule and its negative ion.

**Figure 5 molecules-26-04973-f005:**
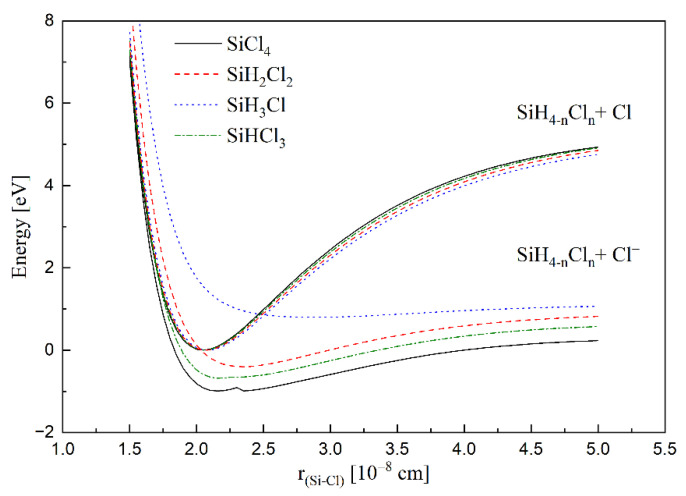
Potential curves for simple chlorosilane molecules and their negative ions.

**Figure 6 molecules-26-04973-f006:**
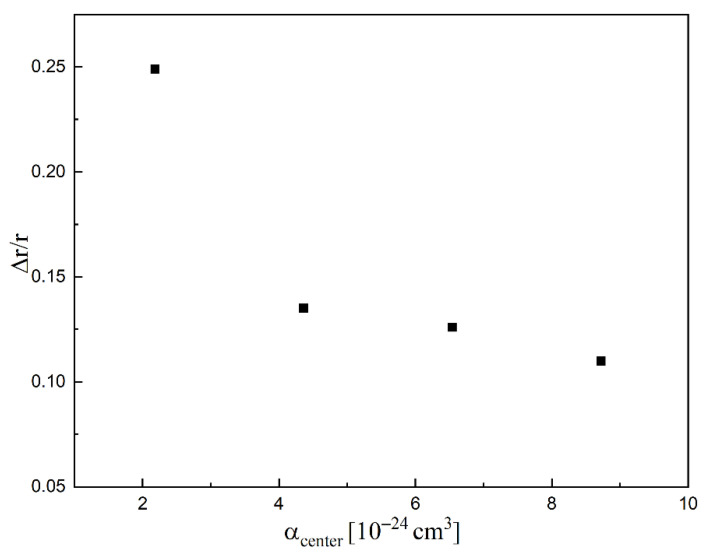
Dependence of the Δ*r/r* on *α*_center_ for SiH_3_Cl, SiH_2_Cl_2_, SiHCl_3_, SiCl_4_.

**Figure 7 molecules-26-04973-f007:**
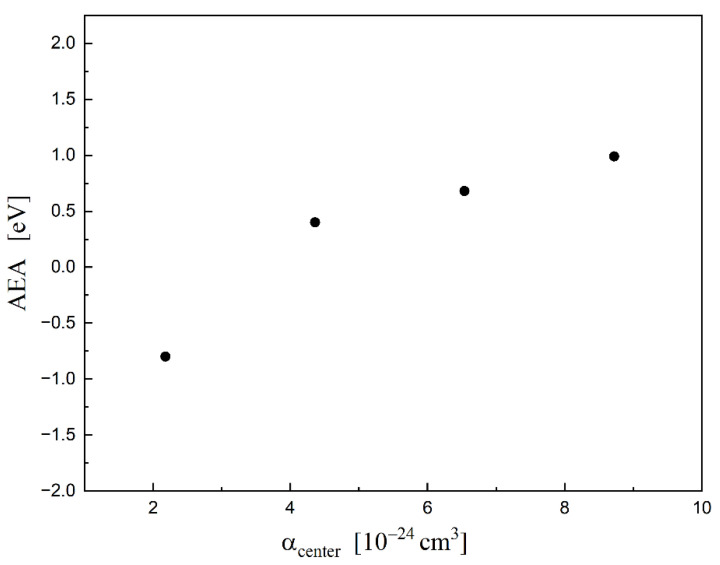
Dependence of the AEA on *α*_center_ for: SiH_3_Cl, SiH_2_Cl_2_, SiHCl_3_, SiCl_4_.

**Figure 8 molecules-26-04973-f008:**
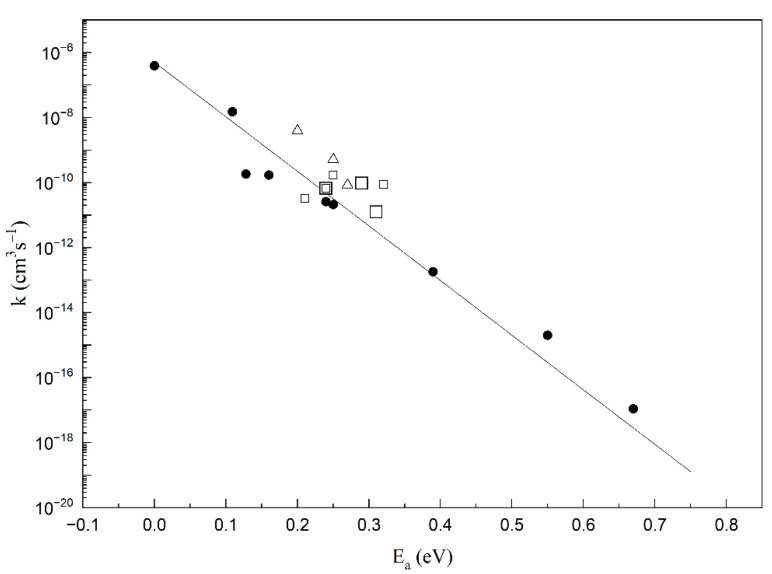
Thermal electron attachment rate coefficients as a function of the activation energies: (Δ)—present data; our previous data for: (□)—chlorinated silanes; (●)—chlorinated alkanes (values from Table 1). Solid line—theoretically obtained values of the rate coefficients for *T* = 298 K considering the electron as the de Broglie wave.

**Table 1 molecules-26-04973-t001:** Thermal electron attachment rate coefficients (*k*_298_) at 298 K and activation energies (*E_a_*) for all of the investigated chlorosilanes and literature data for the selected chlorinated silanes and alkanes. T_range_ corresponds to the temperature range in which *E_a_* were obtained.

Molecules		*k*_298_ (cm^3^·s^−1^)	*E_a_* (eV)	T_range_ (K)
Chlorinated silanes (present data)	SiHCl_3_	(5.18 ± 0.22) × 10^−10^	0.25 ± 0.01	298–368
	SiCl_4_	(3.98 ± 1.8) × 10^−9^	0.20 ± 0.01	298–368
	Si(CH_3_)_2_(CH_2_Cl)Cl	(8.46 ± 0.23) × 10^−11^	0.27 ± 0.01	298–378
Chlorinated silanes	SiH(CH_3_)_2_Cl	(6.72 ± 0.08) × 10^−11^ [13]	0.24 ± 0.01 [13]	298–368
	SiHCH_3_Cl_2_	(16.80 ± 0.06) × 10^−11^ [13]	0.25 ± 0.10 [13]	298–378
	SiCH_3_Cl_3_	(8.72 ± 0.07) × 10^−11^ [13]	0.32 ± 0.02 [13]	298–378
	Si(C_2_H_5_)_3_Cl	(3.27 ± 0.02) × 10^−11^ [13]	0.21 ± 0.01 [13]	298–378
	Si(CH_3_)_3_Cl	(9.56 ± 0.02) × 10^−11^ [14]	0.29 ± 0.01 [14]	298–368
	Si(CH_3_)_2_Cl_2_	(6.62 ± 0.02) × 10^−11^ [14]	0.24 ± 0.01 [14]	298–368
	SiH(CH_3_)_2_(CH_2_Cl)	(1.24 ± 0.05) × 10^−11^ [14]	0.31 ± 0.01 [14]	298–378
Chlorinated alkanes	CH_3_Cl	1.1 × 10^−17^ (300 K) [28]	0.67 [28]	700–1100
	CH_2_Cl_2_	1.8×10^−13^ (300 K) [28]	0.39 [28]	495–973
	CHCl_3_	3.7 × 10^−9^ [29]	0.11 [29]	295–373
	CCl_4_	3.9 × 10^−7^ (300 K) [30]	0.00 [30]	205–590
	CH_3_CH_2_Cl	3.4 × 10^−14^ [31]	-	298
	CH_2_ClCH_2_Cl	2.6 × 10^−11^ [31]	0.24 [32]	298–463
	CHCl_2_CH_3_	2.1 × 10^−11^ [33]	0.25 [32]	298–463
	CH_3_CCl_3_	1.5 × 10^−8^ [34]	0.11 [34]	298–470
	CHCl_2_CH_2_Cl	1.8 × 10^−10^ [34]	0.13 [34]	298–470
	CHCl_2_CHCl_2_	3.5 × 10^−8^ [35]	-	298
	CH_2_ClCCl_3_	3.1 × 10^−7^ [35]	-	298

**Table 2 molecules-26-04973-t002:** Relative changes in the bond length for a negative ion and its neutral, Δ*r/r*, adiabatic electron affinity, AEA, and relative changes in the H-Si-Cl bond angle for a negative ion and its neutral state, Δ*∠/∠*. Calculated using the DFT/B3LYP for chlorine-substituted silanes.

Compound	*r*_(Si-Cl)_ [10^−8^ cm]	*r*_*n*(Si-Cl)_ [10^−8^ cm]	Δ*r/r *^1^	AEA [eV]	*α*_center_ [10^−24^ cm^3^]	∠HSiCl [°]	∠_*n*_HSiCl [°]	Δ*∠/∠* ^2^
SiH_3_Cl	2.079	2.596	0.249	−0.80	2.18	108.63	157.36	0.45
SiH_2_Cl_2_	2.066	2.344	0.135	0.40	4.36	108.28	95.22	0.12
SiHCl_3_	2.055	2.314	0.126	0.68	6.54	109.71	114.32	0.04
SiCl_4_	2.047	2.273	0.110	0.99	8.72	109.47	109.47	0.00

^1^ Δ*r/r* = (*r*_*n*_ − *r*)/*r*, *r*—neutral Si-Cl bond length, r_*n*_—negative ion Si-Cl bond length; ^2^ Δ*∠/∠* = (∠_*n*_ − ∠)/∠.

## Data Availability

The data are available on request from the corresponding author.

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
