# Peer review of "Low Energy Electron Attachment by Some Chlorosilanes"

_molecules, 2021, doi:10.3390/molecules26164973_

Round 1
Reviewer 1 Report
Manuscript report
Title of the manuscript: “Low energy electron attachment by some chlorosilanes”
Special Issue: Cutting Edge Physical Chemistry Research in Europe
Authors: Bartosz Michalczuk, Wiesława Barszczewska, Waldemar Wysocki and Štefan Matejčk
Manuscript number: molecules 1334835
In the manuscript the Authors present an experimental and theoretical study on the thermal electron attachment for chlorosilanes contained molecules. The present work takes part of a series of articles that the authors wrote devoted to determine the rate coefficients and the activation energies for this kind of molecules.
Silanes and chlorosilanes play a primary role in many industrial plasmas as well as in astrophysical contests. The paper, in general, is well written, the theoretical model and the calculation details are correctly presented, the experimental Swarm technique - also known as the pulsed Townsend technique - was properly explained and the results analytically discussed.
I think that the paper itself is up to the standard of the journal ‘molecules’ and it is well aligned with the scope of the journal. I believe that the topic treated by the manuscript is useful for the molecular dynamic community.
For these reasons I strongly suggest for the publication of the paper.
Author Response
Dear Reviewer,
Thank you for the revision and we are more than happy that in your opinion our manuscript is worthy of publication.
Kind regards,
Bartosz Michalczuk
Reviewer 2 Report
Please, see in attached file.

Author Response
Dear Reviewer,
Thank you for the revision and valuable comments. In the attachment please find answers to all comments.
Kind regards,
Bartosz Michalczuk
